# Identification of a NACC1-Regulated Gene Signature Implicated in the Features of Triple-Negative Breast Cancer

**DOI:** 10.3390/biomedicines11041223

**Published:** 2023-04-20

**Authors:** Chrispus M. Ngule, Hami Hemati, Xingcong Ren, Oluwafunminiyi Obaleye, Amos O. Akinyemi, Felix F. Oyelami, Xiaofang Xiong, Jianxun Song, Xia Liu, Jin-Ming Yang

**Affiliations:** 1Department of Toxicology and Cancer Biology, College of Medicine, University of Kentucky, Lexington, KY 40536, USA; 2Department of Microbial Pathogenesis and Immunology, Texas A&M University Health Science Center, Bryan, TX 77807, USA; 3Markey Cancer Center, College of Medicine, University of Kentucky, Lexington, KY 40536, USA; 4Department of Pharmacology and Nutritional Science, College of Medicine, University of Kentucky, Lexington, KY 40536, USA

**Keywords:** tumor progression, cancer stem cells, gene signature, *NACC1*, triple-negative breast cancer

## Abstract

Triple-negative breast cancer (TNBC), characterized by a deficiency in estrogen receptor (ER), progesterone receptor (PR), and human epidermal growth factor receptor2 (HER2), is among the most lethal subtypes of breast cancer (BC). Nevertheless, the molecular determinants that contribute to its malignant phenotypes such as tumor heterogeneity and therapy resistance, remain elusive. In this study, we sought to identify the stemness-associated genes involved in TNBC progression. Using bioinformatics approaches, we found 55 up- and 9 downregulated genes in TNBC. Out of the 55 upregulated genes, a 5 gene-signature (*CDK1*, *EZH2*, *CCNB1*, *CCNA2*, and *AURKA*) involved in cell regeneration was positively correlated with the status of tumor hypoxia and clustered with stemness-associated genes, as recognized by Parametric Gene Set Enrichment Analysis (PGSEA). Enhanced infiltration of immunosuppressive cells was also positively correlated with the expression of these five genes. Moreover, our experiments showed that depletion of the transcriptional co-factor nucleus accumbens-associated protein 1 (NAC1), which is highly expressed in TNBC, reduced the expression of these genes. Thus, the five genes signature identified by this study warrants further exploration as a potential new biomarker of TNBC heterogeneity/stemness characterized by high hypoxia, stemness enrichment, and immune-suppressive tumor microenvironment.

## 1. Introduction

The stratification of breast cancer (BC) into morphologically and molecularly discrete subtypes has improved the overall treatment of patients with this disease [1,2]. TNBC, a subtype of BC with a poor prognosis is characterized by deficiency in the expressions of estrogen receptor (ER), progesterone receptor (PR), and human epidermal growth factor receptor 2 (HER2). Compared with other subtypes of BCs, TNBC is mostly common in women younger than 40 years, women of color, or those with a *BRCA1* mutation. In addition, TNBC is more proliferative and metastatic, and there are few targeted therapies for TNBC. Therefore, patients with this disease are inclined to have a worse prognosis [2,3,4].

Enrichment of breast cancer stem cells (CSCs), a subpopulation of tumor cells is marked by high CD44 expression and low CD24 expression as well as increased *ALDH1A1/3* activity. CSCs intrinsic ability to self-renew, differentiate and repopulate the bulk tumor [5], is believed to contribute substantially to the poor prognosis of TNBC [6]. TNBC tumor heterogeneity driven by CSCs and a lack of effective therapeutic arsenals other than chemotherapy often lead to poor outcomes in patients with TNBC. In recent years, innovations in omics technologies have contributed immense knowledge about TNBC microenvironment heterogeneity [7]. Moreover, identifying the key molecular determinants associated with the tumor heterogeneity and tumor progression in TNBC may help develop novel strategies for treating this malignancy. In this study, we sought to find the critical genes associated with TNBC stemness through analysis of Gene Expression Omnibus (GEO) datasets, The Cancer Genome Atlas (TCGA), and Clinical Proteomics Tumor Analysis Consortium (CPTAC). We also employed the PGSEA algorithm, performed Gene Ontology (GO) analysis, and evaluated protein–protein interaction (PPI) networks to elucidate co-expressed genes with shared biological process terms (BP) and their linkage to stemness pathways in TNBC. Our analyses led to the identification of *CDK1*, *CCNA2*, *CCNB1*, *AURKA,* and *EZH2* as signature genes involved in cancer stemness and progression. Moreover, we found that expressions of these signature genes were affected by NAC1, a transcriptional co-factor belonging to the BTB gene family and possessing an oncologic role [8,9]. The findings reported here may have potential and important implications in the prognosis and treatment of TNBC. 

## 2. Materials and Methods

### 2.1. Identification and Evaluation of Differentially Expressed Genes 

The following keywords: triple-negative breast cancer or TNBC, neoplasm, triple-negative breast cancer, RNA sequencing, TNBC patient samples, or breast cancer patient samples, were used to search in the Gene Expression Omnibus (GEO) database. We identified six TNBC datasets with adjacent normal tissues. The datasets were randomly grouped into training and validation groups. The training group comprised of GSE65194, GSE36295, and GSE38959, while for the validation group, we used GSE21653, GSE20711, and GSE61725 datasets. GEO2R, an online platform using Limma R packages and GEOquery to deal with diverse experimental designs, was used to determine differentially expressed genes in tumors and normal samples. GEO2R analysis [10] was performed with Limma-voom Benjamin and Hochberg and *p* ≤ 0.05 as the settings for all datasets. All the other settings defaulted to GEO2R. We then characterized genes as differentially expressed between TNBC and normal if log2FC was ≥1.5 and the *p*-value was ≤0.05. Common genes in both the training and validation groups were obtained and used for downstream analysis. 

### 2.2. GO Analysis

GO and Kyoto Encyclopedia of Genes and Genomes (KEGG) pathways analysis was performed using Funrich software to determine enriched BP terms and oncogenic processes associated with the differentially expressed genes [11]. Integrated Differential Expression and Pathway Analysis (iDEP), a platform that allows for selection of multiple pathways analysis methods and databases [12,13], was employed to perform parametric analysis of gene set enrichment (PGSEA) [14] to retrieve the genes involved in similar BP and differentially upregulated in TNBC. 

### 2.3. Protein–Protein Interaction (PPI) Network Analysis and Hub Gene Confirmation 

To evaluate the possible PPI of the differentially expressed genes, Searching Tool for Recurring Instances of Neighboring Genes (STRING) [15], Metascape [16], and Cytoscape [15] were used to construct and visualize the networks. UALCAN, a TCGA-based online platform for analysis of gene expression profiles in tumor and normal samples [17,18], was used to determine signature gene expressions in BC subtypes. The TNM plot online platform [19] was employed to compare the identified signature gene expressions in normal, primary, and metastatic samples. Assessment of mRNA expression and copy number alterations was executed using the cBio cancer genomics portal (cbioportal) in TCGA and PanCancer Atlas samples [20]. Immune cells infiltration in correlation with gene expression was analyzed using Timer 2.0, a comprehensive platform for systematic analysis of immune infiltrates in tumors (https://cistrome.shinyapps.io/timer/, accessed on 12 December 2022) [21]. Analysis of protein expression was performed using the Human Protein Atlas database, a repository of tumor and normal IHC stained tissues [22] (https://www.proteinatlas.org, accessed on 12 December 2022), as well as the cBioPortal-Clinical Proteomic Tumor Analysis Consortium (CPTAC) proteomics-pipeline.

### 2.4. Analysis of the Identified Signature Genes Clinical Relevance 

The KM plotter [23] was employed to evaluate the clinical significance of the identified signature genes. Using PAM50, a 50-gene signature classification system [24], we analyzed the implication of the identified genes in TNBC (basal) samples.

### 2.5. In Vitro Expression of Signature Genes in TNBC Cell Lines

To determine protein expression in TNBC cells, we chose a wide range of cell lines across established TNBC subtypes; basal 1 (MDA-MB-468), basal 2 (HCC1806), mesenchymal (BT-459), and mesenchymal stem-like (MDA-MB-231). As controls, MCF10A was used as normal epithelial-like cell lines. Cells were cultured following the ATCC recommendation. DMEM was used to culture MDA-MB-231 and MDA-MB-468; DMEM-F12 was used for MDA-MB-453; and RPMI-1640 for HCC1806 cell lines. MCF10A cells were cultured using mammary epithelial cell growth kit (promo-cell cat#C-21010). The media was supplemented with 10% fetal bovine serum (FBS) and 1% penicillin-streptomycin antibiotic mix. The cells were incubated at 37 °C with 5% CO_2_ and 95% humidity.

### 2.6. Western Blot Analysis

Cells were harvested into a cold PBS buffer. The cells were spun down, and protein extraction was performed using Laemmli buffer supplemented with protease inhibitor. BCA assay kit (Thermo Scientific, #23228, Waltham, MA, USA) was used for protein quantification. Samples were ran in sodium dodecyl sulfate-polyacrylamide gel electrophoresis (SDS-PAGE) at 100 V. Protein blotting was done using immun-Blot^®^ PVDF membrane in transfer buffer (100 V for 75 min). The membranes were blocked with 5% defatted milk for 1 h, and blots were incubated overnight with appropriate primary antibody at 4 °C. Secondary antibody incubation was performed at room temperature for 2 h. TBST buffer was used to wash the membranes for 15 min each before and after adding horseradish peroxidase (HRP)-secondary antibody. Protein expression was visualized using the ChemiDOC^TM^ MP imaging system (BioRad, Hercules, CA, USA).

## 3. Results 

### 3.1. Identification of Highly Altered Genes in TNBC 

We retrieved six datasets from the GEO database and divided them randomly into two groups: training group and validation group. The training group comprised of GSE65194, GSE36295, and GSE38959 datasets, which contained 96 TNBC, 29 normal, and 125 non-TNBC samples; the validation group consisted of GSE21653, GSE61725, and GSE20711 datasets, which had 140 TNBC, 48 normal, and 233 non-TNBC samples (Table 1). 

These datasets were subjected to a systematic analysis as depicted in Figure 1. Differentially expressed genes in each dataset were analyzed using the GEO2R tool (*p* ≤ 0.05 and log2FC ≥ 1.5). The common down- and upregulated genes in each group are shown in Figure 2a and Appendix A. In the training group, we identified 112 common upregulated genes (Figure 2a) and 74 downregulated genes (Appendix A), and in the validation group we observed 67 common upregulated and 23 downregulated genes (Figure 2b and Appendix A). Next, we compared the altered genes in the training group with those in the validation group and found that 64 (55 upregulated and 9 downregulated) genes were common between the training and validation groups (Figure 2c and Appendix A). A heatmap displaying the expression profiles of the common 64 genes in all 6 datasets is shown in Figure 2d. In addition, utilizing the TCGA pan-cancer data, we explored the expression of these 64 genes among TNBC patient samples. Out of the 1084 TCGA BC samples, we analyzed 171 TNBC (basal-like) and 114 normal samples and found that these 64 genes were highly expressed in TNBC compared to normal samples (Figure 2e). 

### 3.2. Correlation of the Highly Expressed Genes with Cancer Progression 

To explore the implication of the highly expressed genes in TNBC formation and progression, we performed a PPI analysis using STRING and Metascape [16]. This analysis revealed possible interactions between the common 55 upregulated genes. The genes interaction network showed 55 nodes, 1359 edges, and an average node degree of 49.4. The average local clustering coefficient of the PPI network was 0.946, with a PPI enrichment *p*-value of 1.0 × 10^−16^ (Figure 3a). The common 55 upregulated genes were further evaluated for their enrichment in the GO-biological process (GO-BP) terms. Intriguingly, we found that these genes were significantly enriched in the pathways associated with cell cycle, PLK1 and Aurora B kinase signaling, DNA replication, FOXM1 transcription factor (TF) network, G2/M checkpoints, and p73 transcription factor network, which are highly dysregulated biological processes in many types of cancer (Figure 3b). Additionally, we found that the regulator of TNBC progression, nuclear transcription factor Y subunit alpha (NFYA), was involved in the regulation of more than 50% of the 55 common upregulated genes (Figure 3c). Our findings imply that the aggressive phenotype of TNBC may be influenced by the overexpression of these genes.

### 3.3. PGSEA Analysis of the TNBC Altered Genes Identifies a Five Gene-Signature with a Putative Role in Tumor Stemness

We next performed a series of bioinformatics assessments and experimental validations to determine the effects of those altered genes on the stemness features of TNBC. Firstly, we performed the Parametric Gene Set Enrichment Analysis (PGSEA) to identify the genes with similar expression patterns and shared common GO-BP terms [14]. This analysis pinpointed several relevant pathways, one of which is the regeneration term that includes *CDK1*, *EZH2*, *CCNB1*, *CCNA2*, and *AURKA* (Figure 3d). We then retrieved the background genes from the regeneration list (157 genes) and delved into the genes likely associated with cancer stemness/cell self-renewal (Appendix A). Among these genes, POU domain-class 5-transcription factor 3 (OCT4), SRY-box transcription factor 2 (SOX2), SRY-box transcription factor 9 (SOX9), Wnt signaling pathway proteins, catenin signaling genes, NOTCH signaling, matrix metallopeptidase 9 (MMP9), Kruppel-like factor 5, and MYC signaling genes, are all implicated in CSCs and self-renewal. In addition, the regeneration term contained several cytokines, such as interleukin-6 (IL-6) and interleukin-10 (IL-10). These analyses suggest that *CDK1*, *EZH2*, *CCNB1*, *CCNA2*, and *AURKA* genes in regeneration terms may have roles in driving the stemness of TNBC. Thus, we selected these five signature genes for further analysis.

The search tool for recurring instances of neighboring genes (STRING) and Cytoscape-cytoHubba validation analyses revealed a highly functional and physical association between these signature genes with paired score degree values greater than 0.64 and a statistically significant enrichment (*p* = 6.17 × 10^−5^). Interaction scores obtained from PPI analysis demonstrated that *CDK1* had the highest connectivity scores with the other proteins (score > 0.9) (Appendix A). Additionally, *CDK1* showed a high interaction with *EZH2* that had a slightly lower interaction with the other three proteins (*EZH2*-Aurora A, *EZH2*-Cyclin B1, and *EZH2*-Cyclin A2). To determine whether these five genes are significantly enriched in cancer-associated GO-biological processes, we performed a functional enrichment assessment using STRING. We found that these genes were not only enriched in cell regeneration, but were also significantly associated with other tumor-related biological processes, including histone phosphorylation, histone modification, and G2/M transition of mitotic cell cycle (FDR < 0.0001) (Appendix A). Furthermore, functional exploration of the five signature genes in the Wikipathways database affirms their roles in tumor-associated pathways, including retinoblastoma tumor regulation, DNA damage response, DNA repair, Ataxia telangiectasia Mutated (ATM) signaling pathway, miRNA regulation of DNA damage response, 5′ adenosine monophosphate-activated protein kinase (AMPK) signaling, cell cycle regulation, and DNA damage response (FDR ≤ 0.02) (Appendix A).

To determine whether the functions of the signature genes were unique to the TNBC phenotype, we compared their expression in TNBC with other BC subtypes at both transcript and protein levels using the TCGA dataset samples. Our analysis found that except *CCNB1*, which its increase in TNBC did not attain a significant level compared to HER2 positive samples, all the other genes were highly expressed in TNBC samples as compared to the normal or other BC subtypes (*p* ≤ 0.05) (Figure 4a). Using the CPTAC Platform samples, we also analyzed the protein expression of the signature genes in TNBC samples (Figure 4b). A significant upregulation of Aurora A, Cyclin B1, *CDK1*, and *EZH2* was observed in TNBC as compared to other types of BCs (O-BCs) (*p* ≤ 0.05). *CCNA2* expression in TNBC was not significantly higher than that in O-BCs. Analysis of the signature genes’ protein expression using the Human Protein Atlas also showed high expressions of these proteins in tumor samples compared to normal samples (Appendix A). Moreover, our Western blot analysis verified the higher expressions of *EZH2*, Aurora A (encoded by *AURKA*) and Cyclin B1 (encoded by *CCNB1*) in MDA-MB-231, and MDA-MB-468 TNBC cells than in non-malignant MCF10A breast cells (Figure 4c). 

Using TCGA dataset, we investigate the potential contributions of the identified signature genes to tumor metastasis using the tumor-normal-metastasis (TNM) plotter platform. The Pearson correlation analysis demonstrated a significantly increased expression of the signature genes in primary and metastatic samples compared to normal samples: *CDK1*, *p* = 4.98 × 10^−56^; *CCNA2*, *p* = 4.42 × 10^−16^; *CCNB1*, *p* = 1.5 × 10^−33^; *AURKA*, *p* = 1.5 × 10^−18^; and *EZH2*, *p* = 6.94 × 10^−7^ (Figure 4d). This analysis not only confirms the increased expression of these genes in TNBC but also supports our hypothesis that these genes have important roles in TNBC tumor progression. The receiver operating characteristic (ROC) analysis also displayed the specificity of these genes to TNBC versus non-TNBC, as illustrated by the high area under the curve (AUC) values: *CCNA2*: 0.857 (*p* < 0.00001); *EZH2*: 0.844 (*p* < 0.00001); *AURKA*: 0.784 (*p* < 0.00001); *CCNB1*: 0.762 (*p* < 0.00001), and *CDK1*: 0.722 (*p* < 0.00001) (Figure 4e). Furthermore, using the KM-Plotter platform, we assessed the clinical implication of the identified signature genes in prognosis of TNBC patients. Among the five signature genes, the expression of *EZH2* and *CCNB1* appeared to highly affect the clinical prognosis of TNBC patients (Figure 4f). Collectively, our datashow that the elevated expression of the identified signature genes in TNBC may substantially contribute to the aggressive phenotype of TNBC, including tumor stemness genes. 

### 3.4. Signature Genes Expression Is Positively Associated with Immunosuppressive Cells Infiltration and Hypoxia Status in TNBC

Recent studies have revealed an association between stemness and immune signatures [25,26,27]. It is believed that CSCs in highly hypoxic regions of the tumor microenvironment (TME) can interact with immune cells, including T-cells, tumor-associated macrophages, and myeloid-derived suppressor cells (MDSCs) [28]. Reciprocally, the immune cells such as MDSCs and Tregs can nurture stemness niche to support CSC survival [27]. We, therefore, ascertain whether the expressions of the signature genes have an association with the infiltration of Tregs and MDSCs in TNBC samples. TNBC-TCGA samples from the Tumor Immune Estimation Resource (TIMER) 2.0. were utilized, and the published guideline for choosing transcriptome quantification methods was followed to determine the influence of gene expression on immunity [29]. We selected the Tumor Immune Dysfunction and Exclusion (TIDE) platform [30] for analysis of MDSCs infiltration and computational pipeline for quantification of the tumor immune contexture from human RNA-seq data (QUANTISEQ) for Tregs. Notably, there was a significant positive correlation between the mRNA expressions of *AURKA*, *CCNA2*, *EZH2*, and *CDK1* with infiltration of MDSCs (*p* < 0.05). Additionally, there was a significant positive correlation between enhanced *AURKA*, *CCNA2*, *EZH2*, and *CDK1* mRNA expression with MDSCs and Tregs infiltration (*p* < 0.05) (Figure 5a). Upon analysis of the cell types infiltrated in the TME, we observed that stromal and immune scores are positively correlated to low tumor purity [31] and that tumor purity is associated with the expression of signature genes, suggesting that the source of signature genes’ mRNA transcriptome was from tumor cells but not from infiltrated immune cells (Figure 5a). Additionally, through explorative analysis of the role of these genes in predicting patients’ response to immune-checkpoint inhibitors (ICIs), including anti-PD1, anti-PD-L1, or anti-CTLA4 antibodies, we found that *EZH2* (*p* < 0.05) and *AURKA* (*p* < 0.05) could better predict outcomes of patients receiving these treatments (Figure 5b–d). Retrieving the immune-TME and TNBC aggression-associated marker genes from the Pan-cancer TCGA-dataset, we found a high positive correlation of the expression of S100A9 and S100A8, two immune suppression marker genes, with *AURKA* and *EZH2* in the TNBC samples. Similarly, a correlation existed between Arginase 1 (ARG1), known to dampen T-cell proliferation, with *EZH2* and *AURKA*. Additionally, high *CCNB1* expression was correlated with elevated expression of KDM5A (a drug resistance marker), while *AURKA* positively correlated with HAVCR2 (TIM3) (a marker for highly exhausted T-cells) (Figure 5e). These analyses imply that the signature genes identified here may have potential roles in modulating immune-TME and the interaction between immune cells and CSCs, contributing to the immunosuppressive phenotype of TNBC.

Oxygen deficiency is one of the features of TME in solid tumors. Oxygen deficiency favors CSCs enrichment and MDSCs accumulation, contributing to an immuno-suppressive environment [32,33]. Thus, we queried whether these signature genes are involved in the modulation of hypoxia. We retrieved the transcriptomic and copy number alterations (CNAs) data from TCGA and quantified tumor hypoxia in TNBC and non-TNBC samples using three independent hypoxia gene signature-based scoring tools: Winter [34], Ragnum [35], and Buffa [36]. The samples were stratified into TNBC (basal-like) and non-TNBC and evaluated for the differences in hypoxic status. We found that the hypoxic level was significantly higher in the TNBC samples than in the non-TNBC (*p* < 0.0001) (Figure 6a). Sorting the samples based on Ragnum score hypoxia intensity and examining the CNAs and mRNA expression of the signature genes uncovered a correlation between mRNA amplification and CNAs with higher hypoxia intensity (Appendix A).

To determine whether the expression patterns of CNAs and mRNA were also exhibited at the protein level, we explored the CPTAC pan-cancer data. A positive correlation between the expression of the five proteins and hypoxia scores status was observed (Appendix A), and all three methods demonstrated similar changes and correlations (correlation coefficient > 0.2) (Appendix A). The proteins that had the highest correlations with the hypoxia scores were *CDK1* (Buffa score: 0.60; Winter score: 0.51; Ragnum score: 0.52) and *EZH2* (Buffa score: 0.52; Winter score: 0.45; Ragnum score: 0.44). Furthermore, the *EZH2*–*CDK1* pair demonstrated a high correlation (correlation coefficient; 0.62), while Aurora A expression was highly correlated with *CDK1* protein levels (correlation coefficient; 0.53) under hypoxia conditions (Appendix A). To determine whether these five signature genes can predict hypoxia status in BC, we stratified the TCGA samples into two groups based on the Buffa score. The samples with positive Buffa scores were grouped as positive responders, while those with zero or negative scores were grouped as non-responders. ROC analysis demonstrated high specificity of these genes in predicting the samples with high hypoxia (Figure 6b). Intriguingly, AUC values of the signature genes matched those of Ragnum (0.88, *p* < 0.00001) and Winter (0.951, *p* < 0.00001) hypoxia prediction scores: *AURKA*, 0.847 (*p* < 0.00001); *CCNB1*, 0.827 (*p* < 0.00001); *CDK1*, 0.806 (*p* < 0.00001); *CCNA2*; 0.835 (*p* < 0.00001); and *EZH2*; 0.812 (*p* < 0.00001). Together, these analyses suggest that the signature genes identified in this study may have important roles in regulating or predicting hypoxic status in TNBC.

### 3.5. Depletion of NACC1 Reduces the Signature Genes Expression

We and others have previously shown that NACC1 plays an oncogenic role in several forms of cancers by supporting cell survival in hypoxic environments. [37]. More recently, we showed that NAC1 (encoded by the NACC1) expression in melanoma cells promotes their evasion from immune surveillance [38]. Thus, we asked whether this transcription co-factor has any effects on those signature genes in TNBC cells. Analysis of the GSE183947 dataset [39] matched normal, primary, and metastatic TNBC samples found a significant upregulation of *NACC1* mRNA level in primary and metastatic tumors compared to normal tissues (Figure 7a). It has been shown that p27 is enriched in less proliferative cells in TNBC [5]. We identified DEGs between p27-low and p27-high TNBC samples in the GSE198713 dataset [5]. Surprisingly, we found that the signature genes were significantly downregulated in p27-high samples (Figure 7b). Consistently, our experiments demonstrated that depletion of NAC1 in MDA-MB-231 TNBC cells marked reduced Aurora A, Cyclin B1, and *EZH2* expressions (Figure 7c). Furthermore, NACC1 expression in patients under ICIs treatment could clearly predict patients OS. Additionally, a combination of NACC1 and *EZH2* expression revealed better patient OS under anti-PD-L1 treatment (Figure 7d). Oncogenic driver gene signature analysis using GSEA on NACC1 depleted cells RNA-seq data showed that the downregulated genes were enriched in PRC2-*EZH2*, KRAS, BC-associated genes, epidermal growth factor receptor (EGFR), AKT signaling, and WNT pathway (Appendix A). Furthermore, our Western analysis on AKT pathway-associated gene expression in the NACC1-depleted tumor cells showed decreased expressions of phospho-PTEN, β-catenin, and cyclin-D1 (Appendix A), which is consistent with the RNA-seq data. The NOTCH pathway represents a major CSC regulator in TNBC [40]. By RNA-seq analysis, we found that NAC1 depletion decreased the expression of NOTCH-associated genes compared with nontargeted controls (Appendix A), suggesting that NAC1 positively regulates the NOTCH pathway. Furthermore, the expression of *EZH2* and *CCNA2* also positively correlated with the expression of NOTCH1(Appendix A). Overall, these findings imply that the upregulation of NAC1 in TNBC may promote tumor progression and hypoxia adaption by affecting the oncogenic and stemness-associated pathways and genes, including the five signature genes identified in this study.

## 4. Discussion

Intra- and inter-tumor heterogeneity, tumor relapse, metastasis, and therapy resistance constitute the major challenges in the treatment of TNBC, and CSCs are an important contributor to these malignant phenotypes [41]. To better understand these problems, we explored the genes that drive the malignant phenotype of TNBC. We mined six GEO datasets and TCGA samples and analyzed the altered gene expressions at both transcriptome and protein levels using PGSEA algorithm, STRING, Metascape, and TIMER2.0. Through these analyses, we identified 55 upregulated and 9 downregulated genes implicated in the regulation of aggressive phenotypes in TNBC. Further exploration led to the identification and validation of *CDK1*, *CCNB1*, *AURKA*, *EZH2*, and *CCNA2*, a five-gene signature with high interactions and potentially associated with tumor stemness, hypoxia enhancement, and TME regulation. Consistently, these five genes are among those reported to be potential prognostic biomarker signatures for hepatocellular carcinoma [42]. 

Protein–protein interactions, as well as gene co-expression, are crucial in the control of various malignant features. Here, we found a close association of the 55 common differentially expressed genes as well as the identified signature genes at the mRNA and protein levels, especially for *CDK1*-*EZH2* pairing. A strong correlation of these genes at the mRNA level may suggest a shared transcription regulation machinery. We further identified NFYA, known to promote tumor progression in TNBC [43], as the most enriched transcription factor targeting over 50% of the upregulated genes. The short-isoform of NFYA transcriptionally amplifies the survival and tumor cell proliferation-associated genes. On the other hand, the long NFYA isoform enhances cell signaling through EMT transition and leads to a more belligerent phenotype of Claudin^low^ basal-like tumors [43]. Similarly, the expression of NYFA and sex determinant region Y are independent prognostic markers in gastric cancer, and that NFYA promotes growth of clear cell renal cell carcinoma (ccRCC) through the regulation of cyclins activity [44,45]. These observations support the hypothesis that upregulation of the signature genes transcription and their protein translation in TNBC play a significant role in tumor progression. 

Recent studies have demonstrated that interaction and phosphorylation of SOX2 by *CDK1* positively regulate stemness in lung cancer [46], and *CDK1* phosphorylation of both HIF-1α and *EZH2* promotes tumor cells survival and proliferation [47,48]. Our analyses show that *CDK1* is highly correlated with *EZH2* as well as other signature genes in TNBC. Additionally, we found that both *CDK1* and *EZH2* are strongly correlated with hypoxia status at transcription, CNAs, and protein translation levels. Under hypoxia, tumor cells can stimulate MDSCs to establish a premetastatic niche that promotes cancer cells aggression [33]. Although tumor cells have the capability to adapt to hypoxia, the hypoxia-related products can deter activation of tumor-infiltrating immune cells, establishing an immunosuppressive environment critical for tumor cell evasion from immune surveillance [33]. *IL-6* is a cytokine that is involved in immunosuppression and its expression is upregulated by *CDK1* activity [49]. Under hypoxia, IL-6 can upregulate HIF1α to enhance CSCs survival under cisplatin therapy [50]. We show here that the upregulation of signature genes (especially *CDK1, EZH2,* and *AURKA*) is correlated with hypoxia in TNBC at the mRNA and protein levels, and that these signature genes are highly expressed in primary and metastatic samples. Increased expression of *EZH2* can enhance TNBC cells invasion and metastasis through suppression of *TIMP2* transcription and subsequent amplification of MMP-2 and MMP-9 activity [51]. The role of hypoxia in regulating immune-TME of TNBC is emerging, and studies have demonstrated the importance of hypoxia in immune–tumor cell interactions that drive tumor progression [52]. The high correlation of the signature genes with tumor hypoxia, stemness status, and increased immune cells infiltration may provide new insights into this issue. Our ROC analysis further corroborates these observations and implies the potential of the identified signature genes in predicting the hypoxia status of TNBC. 

NAC1 expression in melanoma cells was recently reported by us to be essential for immune evasion. [38] and that NACC1 expression negatively regulates the suppressive activity of Tregs [53] and CD8^+^ memory T-cells [54]. The current study suggests that NAC1, likely through the regulation of the identified signature genes, has a role in immunosuppressive TME. The expression of these genes in HCC positively correlates with infiltrations of neutrophils, macrophages, and dendritic cells and could predict HCC patient’s responses to anti-PD-1, anti-PD-L1, and anti-CTLA-4 antibodies [55]. Here, we showed that increased infiltration of MDSCs and Tregs is substantially correlated with high expression of hallmark genes. The infiltration of both Tregs and MDSCs suggests multiple roles of the signature genes in modulating immuno-TME in TNBC. Liu et al. [56] demonstrated that *CCNB1* and *EZH2* together could influence immune cell infiltration and serve as a prognostic hub in prostate cancer. However, our results show that *EZH2* expression but not *CCNB1* is associated with poor prognosis in patients receiving ICIs, and that a combination of *EZH2* and NACC1 may better predict the survival of patients receiving anti-PDL1 treatment. An in vivo study and further validation of these signature genes and their implication in TNBC are ongoing.

In conclusion, we discovered five NAC1-regulated genes implicated in TNBC aggressive phenotypes, including tumor stemness and immune cells infiltration. The expressions of these genes could be further explored as useful prognostic markers of TNBC. The protein products of these genes may also be exploited as potential targets for TNBC therapy.

## Figures and Tables

**Figure 1 biomedicines-11-01223-f001:**
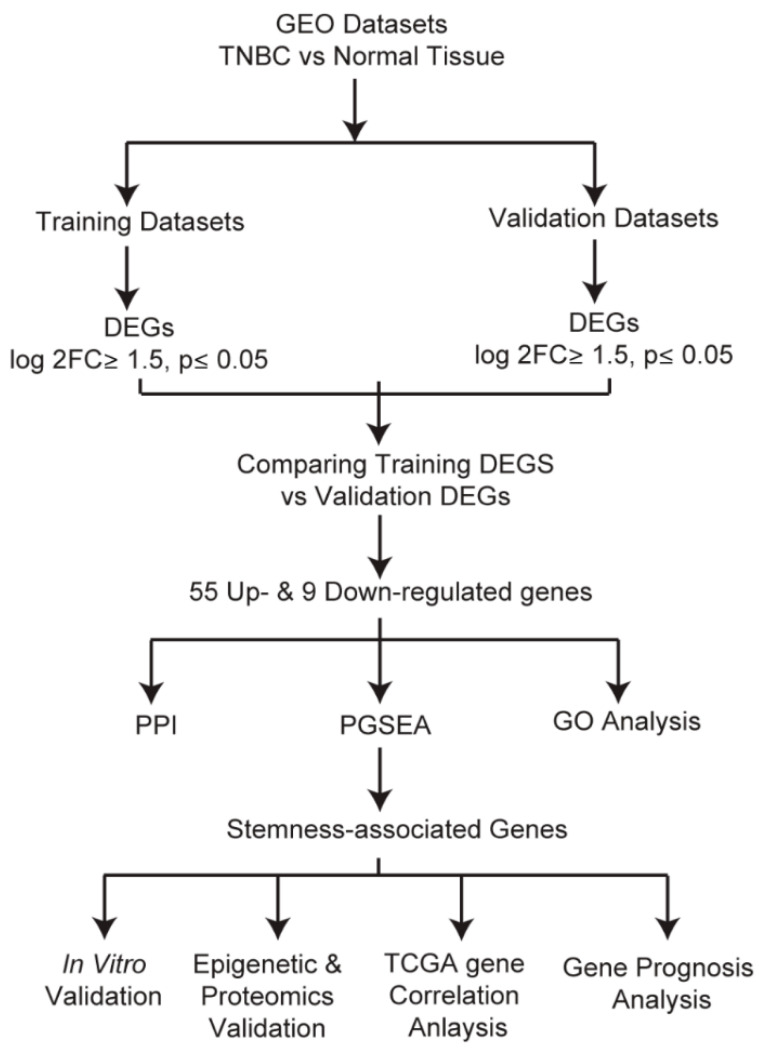
Workflow of the study. In total, six datasets containing TNBC samples were collected from the GEO database and randomly divided into the training or validation groups. Differentially expressed genes were analyzed in the training and validation groups and the common genes found were subjected to further analysis. A total of 64 differentially expressed genes constituting 55 upregulated and 9 downregulated genes were identified. PPI and gene function analysis were performed to evaluate the enriched terms of the common genes. Analyses of genes with the common expression pattern and shared common BP terms were performed using PGSEA. The five genes identified were further validated and evaluated for their potential role in the regulation of CSCs features.

**Figure 2 biomedicines-11-01223-f002:**
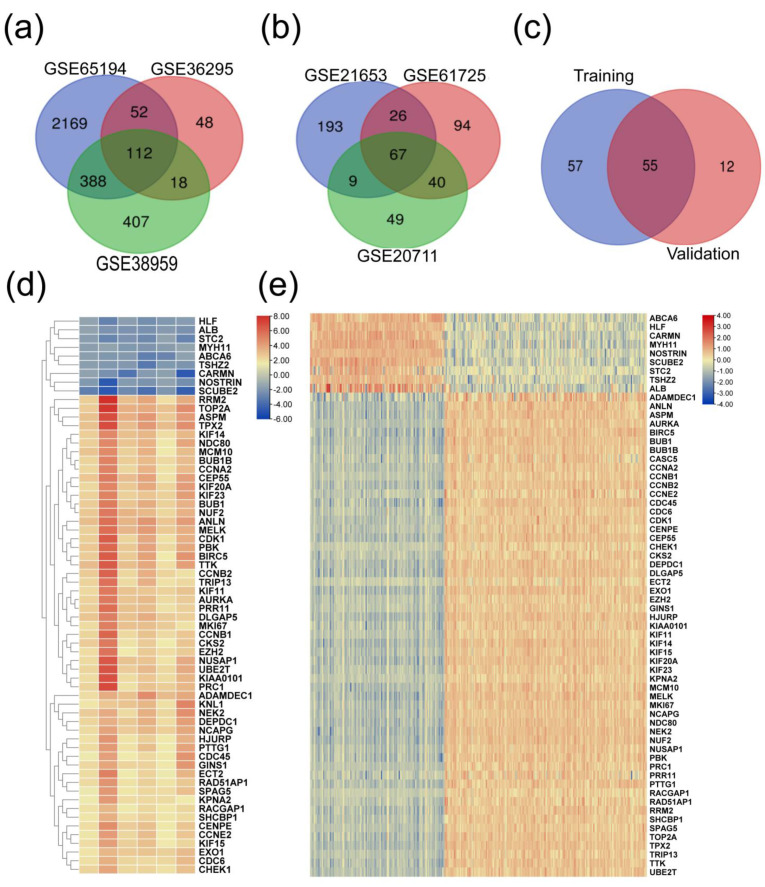
Identification of the genes significantly altered in TNBC using GEO2R. (**a**) The upregulated genes in the training group; (**b**) the upregulated genes in the validation group; and (**c**) the common genes in the training and validation datasets. (*p* ≤ 0.05 and logFC ≥ 1.5). (**d**) Evaluation of the common genes’ expression pattern in the six datasets. (**e**) Confirmation of the expression profile of the common differentially expressed genes in the TCGA pan-cancer TNBC and normal samples.

**Figure 3 biomedicines-11-01223-f003:**
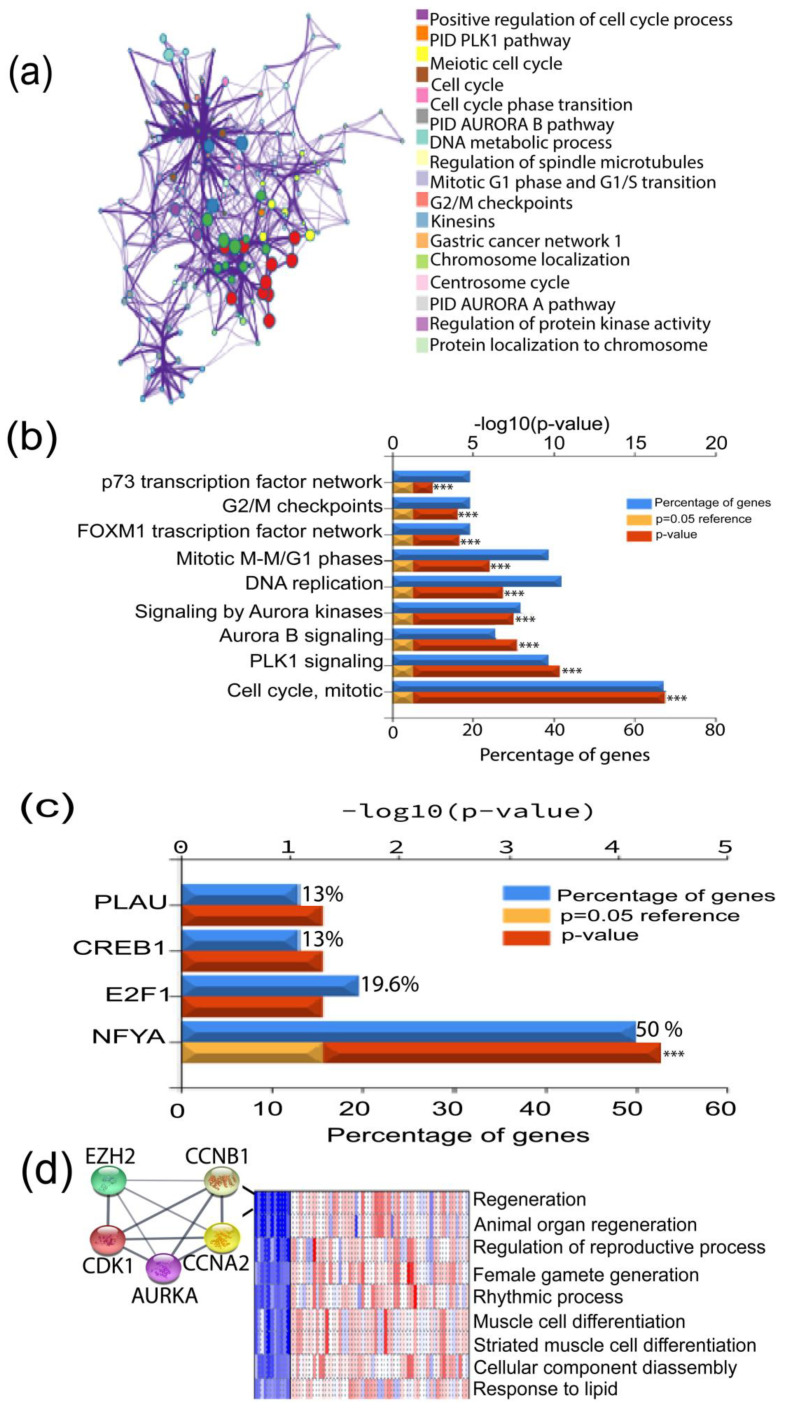
Functional analysis of the 64 common genes identifies a stemness-associated gene signature. (**a**) PPI network of the common differentially expressed genes in the training and validation datasets; (**b**) analysis of differentially expressed gene GO-biological processes pathways enrichment; (**c**) enriched TFs associated with the 55 upregulated differentially expressed genes (*p ≤* 0.001—***); (**d**) PGSEA analysis of the biological processes likely affected by the 55 upregulated and 9 downregulated differentially expressed genes based on their expression patterns in different samples, leading to identification of five genes involved in CSCs phenotype.

**Figure 4 biomedicines-11-01223-f004:**
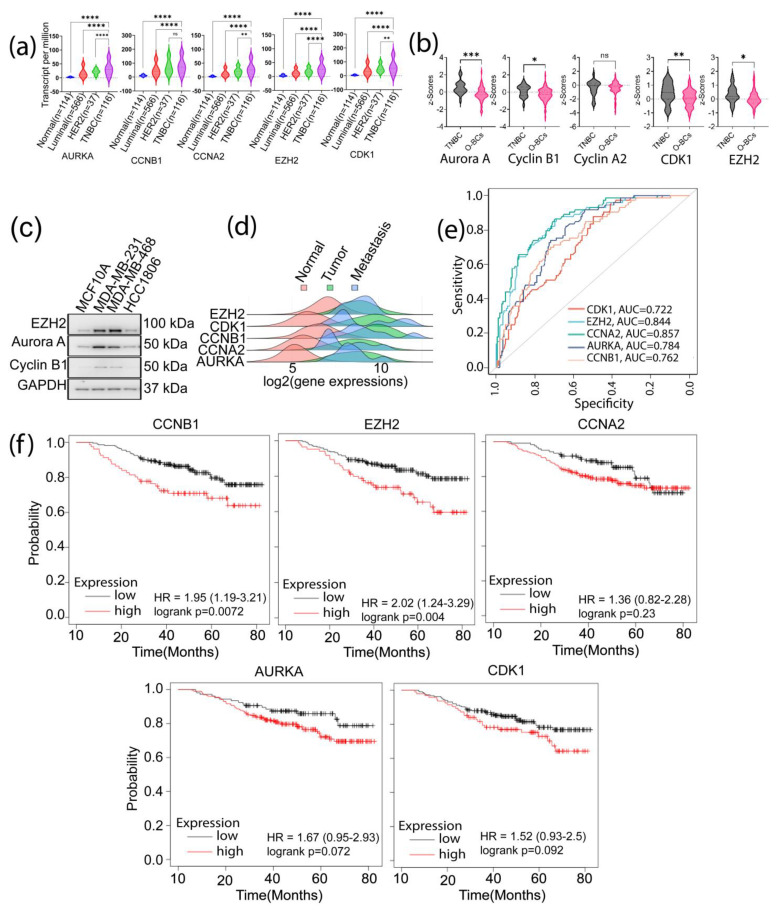
Expression of the five SGs in TNBC, other BCs (O-BCs), and normal samples. (**a**) mRNA expression of the signature genes (SGs) in TNBC, O-BCs and normal samples; (**b**) analysis of the cBioPortal-CPTAC database for protein expression of the SGs in TNBC and other BCs (*p* ≤ 0.05—*, *p* ≤ 0.01—**, *p* ≤ 0.001—***, *p* ≤ 0.0001—****, ns-not significant); (**c**) Western blot analysis of the SGs protein in TNBC cell lines, (**d**) SGs expression in normal, primary, and metastatic samples from TCGA; (**e**) the ROC curve for the SGs, comparing their association to TNBC with normal and other BCs; and (**f**) patients’ overall survival (OS) in relation to signature genes expression. Analysis of clinical relevance of the signature genes in TNBC in relation to basal 50 genes Prediction Analysis of Microarray (PAM50) using the sub-type samples in the KNM plotter platform that comprises 309 TNBC/basal samples.

**Figure 5 biomedicines-11-01223-f005:**
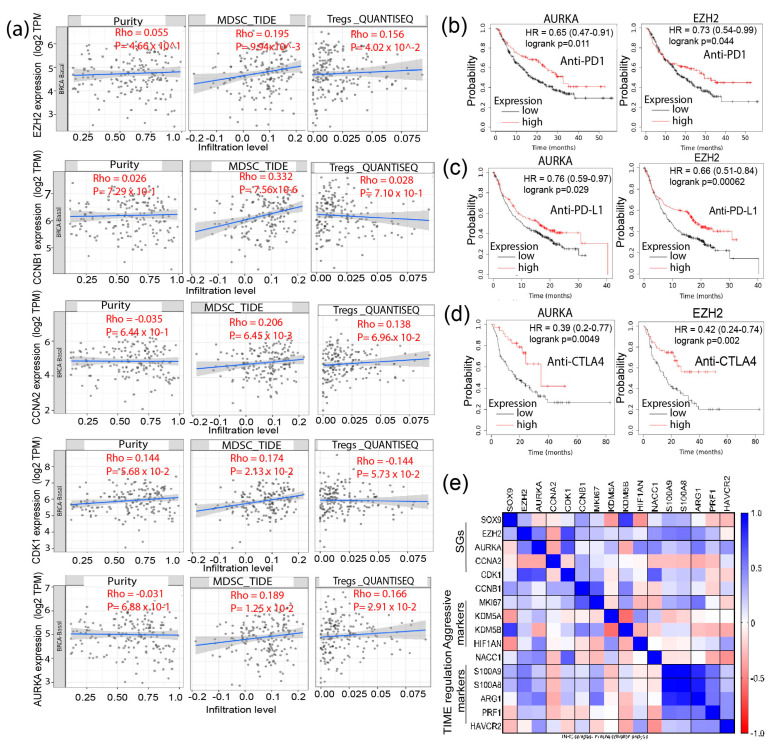
Association of the stemness-associated genes with immune cell infiltration and immunosuppressive markers in TNBC. Correlation of the gene expressions with immune cells infiltration was analyzed using the Timer2.0 online platform. (**a**) Correlation of MDSCs and Tregs infiltration with signature genes (SGs) expression; (**b**) KM plotter analyses of OS of cancer patients treated with anti-PD1 in reference to SGs expression; (**c**) KM plotter analyses of OS of cancer patients receiving anti-PD-L1 treatment in reference to SGs expression; (**d**) KM plotter analyses of OS of cancer patients receiving anti-CTLA4 treatment in reference to SGs expression; and (**e**) a multiple variance analysis of correlation of SGs with immunosuppressive, drug resistance, and proliferative cell markers.

**Figure 6 biomedicines-11-01223-f006:**
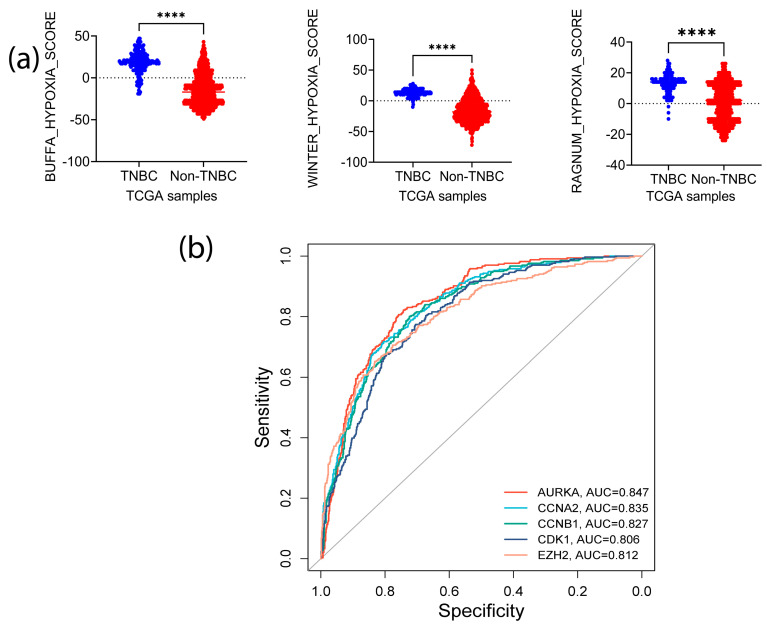
Correlation of the SGs with tumor hypoxic status. (**a**) Hypoxia intensity of TNBC vs. other types of BCs (*p* ≤ 0.0001—****); (**b**) ROC analysis to evaluate the potential of the SGs in the prediction of tumor hypoxia status in TNBC.

**Figure 7 biomedicines-11-01223-f007:**
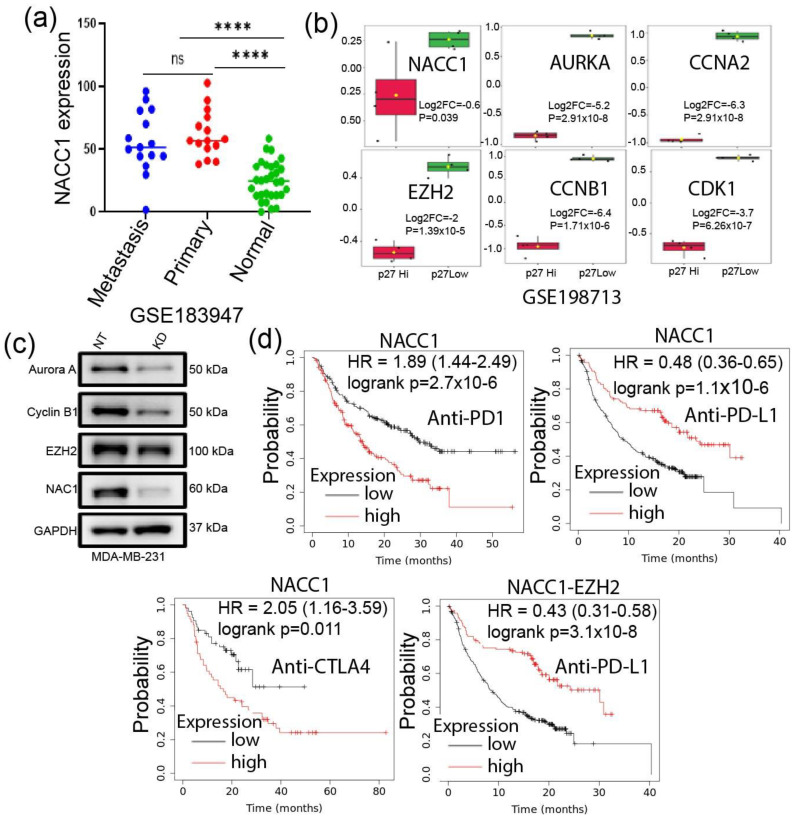
NACC1 depletion reduces the expression of the SGs, and a combination of NACC1 with *EZH2* provides a better prognosis for patients under ICIs therapy. (**a**) NACC1 expression in the matched normal, primary, and metastasis TNBC samples(*p* ≤ 0.0001—****, ns-not significant); (**b**) expression of the five genes in p27 high vs. p27 low samples of GSE198713 datasets; each dots represent a sample, and bars represent median values with range (**c**) expression of Aurora A, Cyclin B1, and *EZH2* in TNBCs cells subjected to depletion of NAC1; and (**d**) overall survival analysis of patients under ICIs treatment in relation to SGs and NACC1 expression.

**Table 1 biomedicines-11-01223-t001:** The dataset compositions.

GEO Datasets	TNBC Samples	Normal	Non-TNBC
Training datasets			
GSE65194	55	11	98
GSE36295	11	5	27
GSE38959	30	13	0
Sum	96	29	125
Validation datasets			
GSE21653	75	29	162
GSE20711	17	2	71
GSE61725	48	17	0
Sum	140	48	233

## Data Availability

The dataset generated for this study is available on request to the corresponding author.

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
