# Peer review of "Identification of a NACC1-Regulated Gene Signature Implicated in the Features of Triple-Negative Breast Cancer"

_biomedicines, 2023, doi:10.3390/biomedicines11041223_

Round 1
Reviewer 1 Report
The manuscript is scientifically sound and well prepared; I have the following minor remarks:
1) The text would benefit from proofreading for typos and grammar, for example I think on page 8 line 4, ("except CCNB1, whose increase in TNBC...") it may be better to use "which" instead of "whose", and so on, some other places in the text as well.
2) Please increase the font on (especially) western blots and some other text on figures, so that it looks unified across the figures, and well readable. Namely figure 4 and figure 7.
Author Response
- “The text would benefit from proofreading for typos and grammar, for example I think on page 8 line 4, ("except CCNB1, whose increase in TNBC...") it may be better to use "which" instead of "whose", and so on, some other places in the text as well.”
Response: We thank the reviewer for such positive comments on our work and have carefully proofread the manuscript and corrected the typos and grammatical errors.
- “Please increase the font on (especially) western blots and some other text on figures, so that it looks unified across the figures, and well readable. Namely figure 4 and figure 7.”
Response: As requested by the reviewer, we have enlarged the font of the text on all the figures including Figure 4 and Figure 7.
Reviewer 2 Report
The manuscript "Identification of a NACC1-regulated gene signature implicated in the features of triple-negative breast cancer" by Liu et al, aims to uncover the role of 5- gene signature motif to better predict treatment outcome by using a number of bioinformatics tools. Triple negative breast cancer (TNBC) is the most aggressive of all Bc subtypes due to lack of expression of surface receptor expression and there is very limited targeted therapies available. Chemotherapy continues to remain the mainstay of treatment. However, clinicians and scientists are still grappling with the question of differential response to established treatment regiment in different individuals. The manuscript offers some compelling analysis in terms of expression of a 5 common gene-signature motif (CDK1, EZH2, CCNB1, CCNA2, and AURKA) to explain the agressive proliferative and metastatic nature of TNBC tumors. The authors with the help of available TCGA and GEO datasets cogently presented that expression of the above 5 genes in TNBC make this subtype of BC more immnosuppressive, hypoxic and metastatic. The significance of this manuscipt is high. The statistical analysis and use of bioinformatics tool by the authors are cutting edge. However, it's difficult to envision how their analysis didn't ascertain Notch signaling or Notch responsive genes. Since Notch is one of the executive pathways in generating cancer stem cells in TNBC. In fact, one of the ways TNBC becomes drug resistant is by upregulating Notch in TNBC stem cells. Moreover, EZH2 collaborates with Notch in TNBC to generate TNBC. Notch also induce IL6 and VEGF is also cooperates with Notch for generating highly hypoxic tumor vasculature. Besides, EZH2 mediated TNBC progression is nothing new,. In fact, there is literature suggesting inhibition of epigenetic modifier, EZH2 can block TNBC progression. The limitations of this paper is lack of originality. The 5 signature genes the authors referred to have already been shown to aggravate TNBC growth, invasion and metastasis individually. Overall, lack of in vivo study limits the enthusiasm of this present manuscript.Author Response
- The statistical analysis and use of bioinformatics tool by the authors are cutting edge. However, it's difficult to envision how their analysis didn't ascertain Notch signaling or Notch responsive genes. Since Notch is one of the executive pathways in generating cancer stem cells in TNBC. In fact, there is literature suggesting inhibition of epigenetic modifier, EZH2 can block TNBC progression.
Response: We thank the reviewer for the insightful comments. Indeed, the NOTCH signaling pathway is critically involved in regulating the stemness in TNBC and its dysregulation contributes to therapy resistance. In the current study, we used unsupervised bioinformatics approaches to identify the statistically significant genes and then bioinformatically narrowed down to a few genes that could serve as potential biomarkers. These approaches, similar to other published studies, may not necessarily shortlist the main genes of a specific signaling pathway to the final list of biomarkers, but the shortlisted genes usually are directly or indirectly associated with those pathways. The signature genes that we identified are directly associated with the known signaling pathways involved in TNBC, including the NOTCH signaling pathway. To directly address this, we have provided additional data (new S.Figure 4c) to show that the signature genes and NAC1 are associated with NOTCH and that depletion of NAC1 leads to a decrease of the expression of genes associated with the NOTCH signaling pathway as demonstrated by PGSEA from RNA sequencing data. We also added the data showing the involvement of these genes, especially EZH2, in NOTCH signaling pathway (new S.Figure 4d) (page 14 lines 23-27).
- The limitation of this paper is lack of originality. The 5 signature genes the authors referred to have already been shown to aggravate TNBC growth, invasion, and metastasis individually. Overall, lack of in vivo study limits the enthusiasm of this present manuscript.
Response: We agree with the reviewer that “The 5 signature genes the authors referred to have already been shown to aggravate TNBC growth, invasion, and metastasis individually”. However, as the reviewer recognized “The statistical analysis and use of bioinformatics tool by the authors are cutting edge”. The current study used these “cutting edge” tools and approaches to analyze over 670 human samples to not only validate those individually identified genes but also demonstrate their potential interaction and theranostic value in treatment of TNBC. Although in vivo study and further validation of these signature genes and their implication in TNBC are ongoing, as this manuscript mainly reports the data and finding of our bioinformatic “dry” study, we thought that those results of “wet” research could be reported in a separate paper. We appreciate the reviewer’s insightful comments and have now discussed this point in the revised manuscript (page 16, lines 17-18).
Round 2
Reviewer 2 Report
The revised manuscript "Identification of a NACC1-regulated gene signature implicated in the features of triple-negative breast cancer" by Liu et al, addressed most of the concerns and questions raised by the reviewers. The authors made sincere effort to make to manuscript better.